# Effects of *Lycium barbarum* L. Polysaccharides on Inflammation and Oxidative Stress Markers in a Pressure Overload-Induced Heart Failure Rat Model

**DOI:** 10.3390/molecules25030466

**Published:** 2020-01-22

**Authors:** Cristina Pop, Cristian Berce, Steliana Ghibu, Iuliu Scurtu, Olga Sorițău, Cezar Login, Béla Kiss, Maria Georgia Ștefan, Ionel Fizeșan, Horațiu Silaghi, Andrei Mocan, Gianina Crișan, Felicia Loghin, Cristina Mogoșan

**Affiliations:** 1Department of Pharmacology, Physiology and Pathophysiology, Faculty of Pharmacy, “Iuliu Hațieganu” University of Medicine and Pharmacy, 400349 Cluj-Napoca, Romania; cristina.pop.farmacologie@gmail.com (C.P.); stelianaghibu@yahoo.com (S.G.); cmogosan@umfcluj.ro (C.M.); 2Experimental Medicine and Practical Skills Center, “Iuliu Hațieganu” University of Medicine and Pharmacy, 400349 Cluj-Napoca, Romania; cristi.berce@gmail.com; 3Department of Pathology and Clinical Medicine, Faculty of Veterinary Medicine, University of Agricultural Sciences and Veterinary Medicine, 400372 Cluj-Napoca, Romania; icscurtu@gmail.com; 4Department of Hematology, Institute of Oncology “Prof. dr. Ion Chiricuță”, 400015 Cluj-Napoca, Romania; olgasoritau@yahoo.com; 5Department of Physiology, Faculty of Medicine, “Iuliu Hațieganu” University of Medicine and Pharmacy, 400012 Cluj-Napoca, Romania; cezarlogin@gmail.com; 6Department of Toxicology, Faculty of Pharmacy, “Iuliu Hațieganu” University of Medicine and Pharmacy, 400349 Cluj-Napoca, Romania; kbela@umfcluj.ro (B.K.); ionel.fizesan@umfcluj.ro (I.F.);; 7Surgical Clinic, Department of Surgery, Faculty of Medicine, “Iuliu Hațieganu” University of Medicine and Pharmacy, 400012 Cluj-Napoca, Romania; 8Department of Pharmaceutical botany, Faculty of Pharmacy, “Iuliu Hațieganu” University of Medicine and Pharmacy, 400337 Cluj-Napoca, Romania; gcrisan@umfcluj.ro

**Keywords:** heart failure, abdominal aortic banding, anti-inflammatory, antioxidant, *L. barbarum* polysaccharides

## Abstract

Despite recent advances in disease management and prevention, heart failure (HF) prevalence is still high. Hypertension, inflammation and oxidative stress are being investigated as important causative processes in HF. *L. barbarum* L. polysaccharides (LBPs) are widely used for their anti-inflammatory and antioxidant properties. Thus, the aim of the present study was to evaluate the effects of LBPs on inflammation and oxidative stress markers in a pressure overload-induced HF rat model, surgically induced by abdominal aorta banding in Wistar rats (AAB) (*n* = 28). Also, control rats (*n* = 10) were subjected to a sham operation. After echocardiographic confirmation of HF (week 24), AAB rats were divided into three groups: rats treated with LBPs for 12 weeks: 100 mg/kg body weight /day (AAB_100, *n* = 9), 200 mg/kg body weight /day (AAB_200, *n* = 7) and no-treatment group (control AAB, *n* = 12). After 12 weeks of treatment with LBPs, the decline of cardiac function was prevented compared to the control AAB rats. Treatment with 200 mg/kg body weight /day LBPs significantly reduced the inflammation as seen by cytokine levels (IL-6 and TNF-α) and the plasma lipid peroxidation, as seen by malondialdehyde levels. These results suggest that LBPs present anti-inflammatory and antioxidant effects with utility in a HF animal model and encourage further investigation of the cardioprotective effects of these polysaccharides.

## 1. Introduction

Heart failure (HF) is a clinical syndrome caused by structural and/or functional cardiac modifications resulting in impairment of the ventricles to fill with or eject blood [1]. HF comprises three main subtypes: HF with normal left ventricular ejection fraction (LVEF) (LVEF ≥ 50%), also called HF with preserved ejection fraction (HFpEF), HF with reduced LVEF (LVEF < 40%), also called HFrEF and HF with mid-range LVEF (LVEF = 40%–49%), also called HFmrEF [1]. In general, HF is a major cause of morbidity and mortality in the developed world, with a high prevalence in the general population (1%–2%) [1]. HF prognosis is concerning, with around 2%–17% of patients dying during the first hospital admission and more than 50% within the first five years [2]. Moreover, HF is a complex disease with a multifactorial etiology, arterial hypertension being among the most prevalent causes for all HF subtypes [1]. One animal model with clinical relevance in the study of HF is the abdominal aortic banding (AAB) in rats. AAB closely mimics the development of HF in humans as a cause of hypertension and can be used to study new therapeutic approaches in specific states of the disease [3]. In this model, pressure overload gradually increases in intensity as the rat matures and leads to structural and functional abnormalities characteristic for HF [4]. Also, inflammation, immune activation and oxidative stress have been identified as important causative processes in HF development and progression. Independently of the etiology, the pathophysiologic mechanisms that influence immune activation can be identical [5]. Extensive activation of renin–angiotensin–aldosterone system in HF is a major pathophysiologic mechanism in HF and recently, aldosterone has been found to increase the expression of proinflammatory cytokines in macrophages, whereas natriuretic peptides have been identified to display anti-inflammatory effects in vivo [5]. Excess mitochondrial production of reactive oxygen species, in comparison to antioxidant defense, has been observed as an important step in the pathophysiology of cardiac remodeling and HF [6]. Thus, treatment with antioxidant complexes could be an interesting possible option to decrease HF progression and offer cardioprotection.

*Lycium barbarum* L. (*L. barbarum*, known as goji berry or wolfberry) is a member of the Solanaceae family, well known in traditional herbal medicine, especially in China as a renowned Yin strengthening agent [7]. Scientific investigations have focused on antioxidative and immunomodulatory effects in the context of atherosclerosis, neurodegenerative diseases and diabetes [7]. The therapeutic effects of *L. barbarum* have been attributed to its active polysaccharide complex (LBP), comprising six monosaccharides (galactose, rhamnose, glucose, mannose, arabinose and xylose). Cardioprotective effects of LBPs have already been observed in models of ischemia-reperfusion myocardial damage [8], doxorubicin-induced cardiotoxicity [9,10,11] and transgenic models, such as mice with microRNA-1 overexpression [12].

Thus, the aim of our research was to evaluate the effects of LBPs in a pressure overload-induced HF model by echocardiography and analysis of inflammation and oxidative stress markers in rats.

## 2. Results

### 2.1. Body Weight

Body weight (BW) was monitored weekly throughout the experiment. At 24 weeks post-surgery, all three ligatured groups exhibited a lower BW (approximately 27% lower) compared to the control group, trend which was statistically significant (*p* < 0.01) at 36 weeks (Figure 1). Also, it was noted that treatment with the high dose of LBPs (200 mg/kg body weight) inhibited BW decline; BW in the AAB_200 group at 36 weeks post-ligature was approximately 10% higher compared to AAB group (Figure 1).

Also, the heart-to-body weight ratio (heart/BW) was significantly (*p* < 0.05) increased in the three AAB rat groups compared to control group (Table 1), suggesting cardiac hypertrophy. LBPs did not influence this ratio significantly, although in AAB_200 group the heart/BW ratio was approximately 7% lower than in the other two ligatured groups. Moreover, the kidney-to-body weight ratio (kidney/BW) was also found to be significantly higher in the three ligatured groups suggesting kidney hypertrophy after ligature placement above the renal arteries (Table 1).

Lung-to-body weight (lung/BW) and liver-to-body weight (liver/BW) ratios were not found to be significantly different among the groups (Table 1).

### 2.2. Effects of LBPs on Echocardiographic Parameters

At 24 weeks post-ligature placement, anterior and posterior wall thickness (AWT and PWT) were increased in all three ligatured groups compared to control group (Figure 2a,b).

Additionally, left ventricular end-diastolic and systolic diameters (LVEDD and LVESD) were significantly increased in the ligatured rats (Figure 2c,d). Also, left ventricular (LV) mass was significantly increased at 24 weeks post-ligature placement in the AAB groups compared to controls (Figure 2e). Moreover, at week 24, a significant decrease in ejection fraction (EF) was noted for all AAB groups compared to control rats (75.63 ± 3.29 vs. 45.58 ± 1.96, *p* < 0.05). Modifications in the echocardiographic parameters at 24 weeks confirmed the development of HFmrEF and represented the beginning of the treatment period. At 36 weeks post-ligature, structural and functional parameters in the control AAB rats (no treatment ligatured rats) declined further compared to the previous timepoint (week 24): EF was approximately 10% lower, LVEDD was approximately 12% higher and LV mass was 8.27% higher (Figure 2c,e,f). At the end of the study, no statistically significant differences were noted after 12 weeks of treatment with LBPs in structural and functional cardiac parameters (Figure 2). However, LVEDD was 7.82% lower in AAB_200 group compared to the control AAB group and EF was also 9.73% higher in the treatment group (AAB_200) compared to the control AAB group. Also, compared to week 24, at week 36, the LVEDD and EF were not significantly modified in the treatment groups (Figure 2c,f).

### 2.3. Effects of LBPs on Hematology Parameters

In our experimental conditions, the hematology parameters were influenced neither by pressure-overload and cardiac failure development nor by the treatment used (Table 2).

### 2.4. Effects of LBPs on Plasma Cytokines Levels

Plasma cytokines (TNF-α and IL-6) levels at 36 weeks post-surgery were significantly higher in the ligatured rats (AAB, AAB_100, and AAB_200) in comparison to the control group. Plasma TNF-α was significantly lower in the AAB_200 group compared to AAB and AAB_100 groups. Similarly, IL-6 levels in the AAB_200 group were significantly lower compared to AAB and AAB_100 groups (Figure 3).

### 2.5. Effects of LBPs on Plasma Lipid Peroxidation

Malondialdehyde (MDA) levels, the end-product of lipid peroxidation, evaluated at 36 weeks post-surgery were significantly higher in all ligatured rats (AAB, AAB_100, and AAB_200) in comparison to the control group (Figure 4). However, in rats treated with 200 mg/kg body weight LBPs significantly lower levels of MDA were recorded, compared to the other two ligatured groups: AAB and AAB_100 groups (Figure 4).

## 3. Discussion

*Lycium barbarum* L. (*L. barbarum*) has been used in traditional Chinese medicine for a long time. Polysaccharides extracted from *L. barbarum* comprise 6 sugars (galactose, rhamnose, glucose, mannose, arabinose and xylose; molar ratios 2.43, 4.22, 1.38, 0.95, 1 and 0.38, respectively), with furan and pyran rings and alpha and beta anomeric configurations [13]. LBPs are important active constituents that have demonstrated pharmacological functions including antioxidative, immunomodulatory, antitumor, antiaging, neuroprotective, hypoglycemic and hypolipidemic effects [7]. However, there are limited reports regarding effects of LBPs in the context of cardiovascular diseases. Of all cardiovascular diseases, HF has a high prevalence, with over 26 million people diagnosed worldwide. Despite advances in disease management, a diagnosis of HF carries significant risk of morbidity and mortality [14]. Thus, the need for new therapeutic approaches in HF is still high and requires both the use of animal models that reproduce the human disease and also the testing of remedies with potential beneficial effects.

In the present study, HF was induced by banding of the descending abdominal aorta. This procedure induced a gradual increase in left ventricular after-load pressure over time, as previously described [4]. This particular model is similar to the pathological evolution from hypertension to HF in humans; thus, this model is useful for the investigation of the effects of different therapeutic approaches in this particular context [15,16]. So far, studies showed that at about 18 weeks after abdominal aortic banding, rats exhibit high pressure-overload, which in time induced structural and functional abnormalities indicative of HF [3,4,17]. The present study confirms that at approximately 24 weeks post-surgery, AWT, PWT and LV mass increase, suggesting LV hypertrophy. Also, at this timepoint, LVESD and LVEDD increase and EF decreases, suggesting LV dilatation and systolic dysfunction. All these modifications are indicative of HFmrEF. After echocardiographic confirmation of HF, treatment with LBPs was initiated and terminated 12 weeks thereafter, at 36 weeks of the study. The aim of our study was to evaluate the effects of LBPs on inflammation and oxidative stress markers after the onset of the cardiac pathology and not the preventive effects. At the end of the treatment period (week 36), in the no treatment group (control AAB group) EF, an important marker of systolic function, decreased with ~10% compared to week 24, whereas in the treatment groups (AAB_100 and AAB_200), EF remained constant, suggesting possible prevention of cardiac structural and functional alteration. Similarly, in a mouse model of structural remodeling and cardiac contractile dysfunction induced by over-expression of microRNA-1, treatment with LBPs for two months resulted in cardiac contractile function improvement, more precisely cardiac output (CO), end-systolic pressure (ESP), end-systolic volume (ESV) and maximum derivative of change in systolic volume (dP/dt_max_) improvement [12].

Also, our results show marked increases in circulating cytokines, including TNF-α and IL-6 and oxidative stress markers, such as malondialdehyde in AAB rats compared to controls. These results confirm preexisting clinical and preclinical studies that demonstrated the prevalence of oxidative stress, inflammation and immune system activation during HF [2,6,18].

More specifically, similar to our findings, LV dilation and systolic dysfunction were found in other studies to be associated with a marked increase in circulating cytokines, including TNF-α and IL-6 [4,19]. The present study shows a significant decrease of TNF-α and IL-6 after 12 weeks of treatment with 200 mg/kg bw/day LBPs. On the other hand, the lack of modifications in white blood cell count in AAB rats compared to control rats could signify that the increase in plasma cytokines does not originate from a systemic inflammatory state induced by white blood cells, but from more complex cardiac cellular signaling involving cytokines. To the best of our knowledge, our results represent the first reporting of LBPs effects on proinflammatory cytokines levels in a cardiovascular disease context. Such effects have been demonstrated before for LBPs, but mostly related to cancer immunotherapy [7,20].

Moreover, our results show that HFmrEF is associated with significant oxidative stress expressed by significantly increased plasma MDA levels, the end-product of lipid peroxidation, in AAB rats compared to controls. Studies show that because cardiomyocytes have lower levels of antioxidant enzymes, such as superoxide dismutase (SOD) and glutathione-peroxidase (GSH-Px), the heart is more prone to oxidative stress [21]. Twelve weeks of treatment with LBPs demonstrated a significant beneficial reducing effect over MDA levels for the 200 mg/kg bw/day dose of LBPs. The powerful antioxidant effect of *L. barbarum* has been used for a long time in traditional Chinese medicine in the context of age-related diseases and has been demonstrated to derive primarily from LBPs [7]. The antioxidant effects of LBPs have been detected in various in vitro and in vivo assays. In vitro antioxidant activity has been observed in the β-carotene/linoleic acid assay, the scavenging activity towards the superoxide anion and reducing capacity and the inhibition of AAPH [2,2′-azobis(2-amidinopropane)dihydrochloride]-induced hemolysis assay [7]. In vivo antioxidant effects were observed in the streptozotocine-induced diabetes in rats, heat-induced damages in rat testes, high cholesterol diet-fed rabbits and mice, ischemia-reperfusion damage in rat hearts and doxorubicine-induced cardiotoxicity [7,8,9]. In the doxorubicine model, MDA levels were high, whereas SOD and GSH-Px levels were lower in the doxorubicine-treated rats. Cardiotoxicity originates from mitochondrion damage, with cytochrome C release into the cytoplasm and caspase-3 induction of apoptosis. The beneficial effects of LBPs are thought to partially come from the induction of the expression of antiapoptotic protein Bcl-2, with this effect contributing to their cardioprotective potential by attenuating doxycycline-induced cardiac myofibrillar disarrangement in rats [9].

It is possible that together, the antioxidant and immunomodulatory effects have a positive effect on systolic function and lead to cardioprotection. This cardioprotective potential of LBPs and the specific pharmacologic mechanisms involved should be further investigated.

## 4. Materials and Methods

### 4.1. Animals and Experimental Protocol

Four-week-old male Wistar rats (*n* = 38; average body weight 150 g) were provided by the Experimental Medicine and Practical Skills Center of the “Iuliu Hațieganu” University of Medicine and Pharmacy in Cluj-Napoca, Romania. Rats were kept in standard laboratory conditions, with 12-h light/dark cycles, a constant environmental temperature and received standard pellets and water ad libitum.

All experiments and procedures were performed in conformance with the Code of Practice for the Housing and Care of Animals Used in Scientific Procedures [22] and with the Universities Federation for Animal Welfare guidelines. The experimental protocol was approved by the Ethical Committee of the “Iuliu Hațieganu” University of Medicine and Pharmacy in Cluj-Napoca, Romania.

Rats were divided into two main groups: rats that were subjected to sham operation (control rats, *n* = 10) and rats that were subjected to abdominal aortic banding (AAB rats, *n* = 28). Abdominal aortic banding was performed as previously described [4]. Briefly, after anesthesia (50 mg/kg ketamine and 10 mg/kg xylazine, i.m.), a suture line (3–0 silk) was placed on the abdominal aorta, above the renal arteries. In order to obtain a constant reduction of blood flow, a 23G (0.6 mm diameter) blunt needle was placed alongside the aorta and the suture line was tied around the aorta and the needle. Afterwards, the needle was removed and the abdominal cavity was closed. For the control group the abdomen was opened and closed, without placement of aortic banding.

Subsequently, AAB rats were subdivided into three groups: AAB control group (*n* = 12) and two other groups treated with different doses of *L. barbarum* polysaccharides (LBPs): AAB_100 (*n* = 7) and AAB_200 (*n* = 9).

A commercially available LBPs extract was used, standardized to 60% polysaccharides. LBPs were mixed in drinking water and administered orally to the AAB_100 group (100 mg/kg body weight/day) and the AAB_200 group (200 mg/kg body weight/day). The control AAB group did not receive any treatment. The total treatment period was 12 weeks starting with week 24 of the study. Body weight (BW) was monitored every two weeks during the experiment. Afterwards, rats were sacrificed at week 36 of the study. Organs (heart, lungs, kidneys and liver) were harvested and weighed. Blood was collected on EDTA from the retro-orbital sinus. Plasma was separated by centrifugation (4000 rpm, RT, 6 min) and samples were stored at −80 °C until use.

### 4.2. Echocardiography Measurements

Rats were anesthetized (30 mg/kg ketamine and 0.5 mg/kg xylazine, i.m.) and transthoracic echocardiography was performed in the supine or left lateral position, every 12 weeks. A commercially available echocardiograph, equipped with a 7.5 MHz electric transducer (Ultrasonix, Boston, MA, USA), was used to acquire two-dimensional echocardiography images at the mid-papillary muscle level. Structures were manually measured by the same observer, who was an expert in animal cardiology, using the leading-edge method of the American Society of Echocardiography [23]. The main measured parameters included: left ventricular anterior wall thickness (AWT), left ventricular posterior wall thickness (PWT), LV end-diastolic and end-systolic diameters (LVEDd and LVEDs), LV mass and ejection fraction (EF).

### 4.3. Complete Blood Count

Complete blood count (CBC) was performed on fresh whole blood, before centrifugation, using Abacus Junior Vet analyzer (Diatron GmbH, Austria). The analyzer provided results for: white blood cells count (WBC), lymphocytes count (LYM), other leucocytes (except lymphocytes and granulocytes) count (MID), granulocytes count (GRA), lymphocytes percentage (LY), other leucocytes percentage (MI), granulocytes percentage (GR), red blood cells count (RBC), hemoglobin (HGB), hematocrit (HCT), mean corpuscular volume (MCV), mean corpuscular hemoglobin (MCH), mean corpuscular hemoglobin concentration (MCHC), red cell distribution width (RDWc), platelets (PLT), platelet proportion in plasma (PCT), mean platelet volume (MPV), platelet distribution width (PDWc).

### 4.4. Proinflammatory Plasma Cytokines

The levels of interleukin-6 (IL-6) and tumor necrosis factor-α (TNF-α) in plasma were analyzed using enzyme linked immunosorbent assays (ELISA kits) (ThermoFisher Scientific, Waltham, MA USA) according to the manufacturer’s instructions.

### 4.5. Plasma Lipid Peroxidation

Malondialdehide (MDA) analysis was performed using a Waters Acquity UPLC system coupled with Waters Acquity photo diode array detector (Waters, Milford, MA, USA), as previously described [24]. Briefly, in order to quantify total MDA (free and protein bound) samples were submitted to a hydrolysis step at 60 °C in a water bath, in the presence of NaOH. After removing proteins with perchloric acid, MDA was derivatized with 2,4-dinitrophenylhydrazine. The derivatization product was extracted in n-hexane, followed by the evaporation of the organic layer. The residue was dissolved in mobile phase and subjected to UPLC–PDA analysis. Chromatographic separation was achieved on a BEH C18 column (50 mm × 2.1 mm i.d., 1.7 mm) from Waters (Waters, Milford, MA, USA), with a mixture of 1% formic acid/acetonitrile as the mobile phase (0.3 mL/min), and gradient elution. The total chromatographic runtime was 7.5 min. The absorbance of the eluent was monitored at 301 nm. Data acquisition and processing were performed using Empower 2 software (Waters, Milford, MA, USA).

### 4.6. Data Analysis

All data are expressed as mean ±SD (standard deviation). Statistical analysis was performed with MedCalc version 19.0.6 and GraphPad Prism 5. One-way ANOVA and t-students tests were applied. Differences were considered statistically significant at *p* < 0.05. Post-hoc analysis was applied for statistical comparisons among groups.

## 5. Conclusions

In the present study, the effects of *L. barbarum* polysaccharides (LBPs) on inflammation and oxidative stress markers were tested in an experimental model of HFmrEF. LBPs presented beneficial effects over systolic function by maintaining the LVEDD and EF after 12 weeks of treatment. Also, 200 mg/kg bw/day dose of LBPs exhibited antioxidant and anti-inflammatory effects. Lastly, rats treated with 200 mg/kg bw/day dose of LBPs had a tendency to be less thin and have a lower heart/BW proportion. The results show that it is possible to obtain significant antioxidant and anti-inflammatory effects using minimal doses of LBPs in a 12-week treatment period. The results obtained in our experimental conditions are promising; thus, the cardioprotective potential of LBPs should be further investigated.

## Figures and Tables

**Figure 1 molecules-25-00466-f001:**
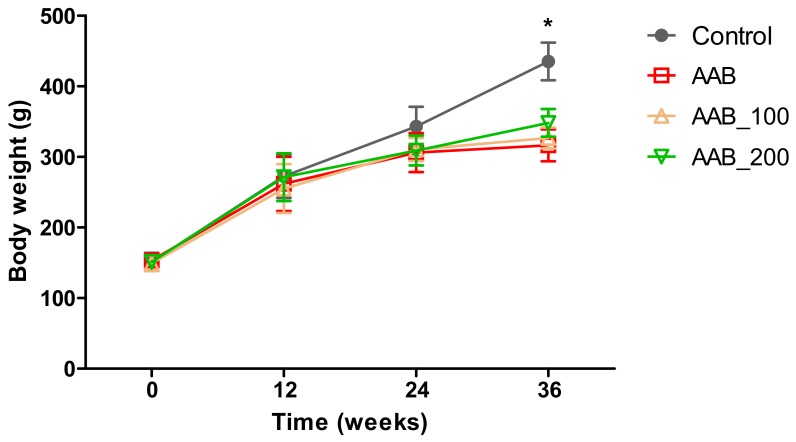
Evolution of rats’ body weight during the experiment. * *p* < 0.01 control vs AAB groups.

**Figure 2 molecules-25-00466-f002:**
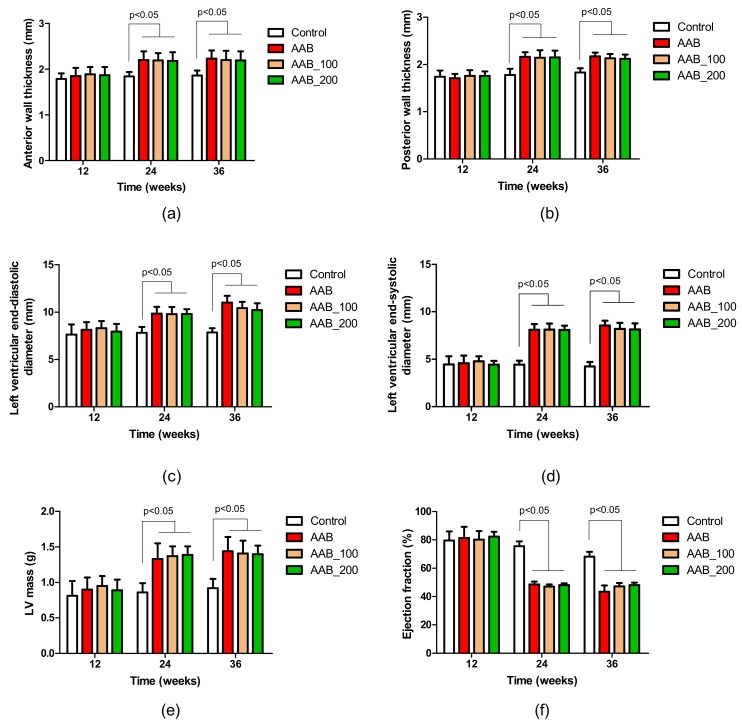
Evolution of echocardiographic parameters. (**a**) Anterior wall thickness (AWT); (**b**) Posterior wall thickness (PWT); (**c**) Left ventricular end-diastolic diameter (LVEDD); (**d**) Left ventricular end-systolic diameter (LVESD); (**e**) Left ventricular (LV) mass; (**f**) Ejection fraction.

**Figure 3 molecules-25-00466-f003:**
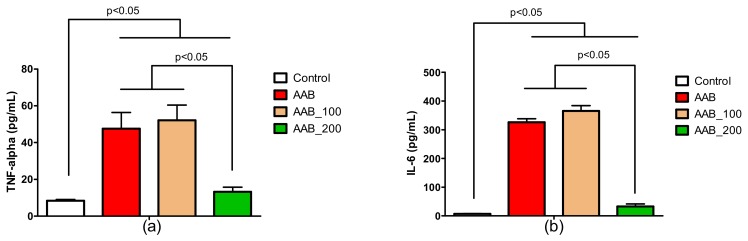
Effects of LBPs on cytokines levels at week 36: (**a**) plasma TNF-α levels; (**b**) plasma IL-6 levels.

**Figure 4 molecules-25-00466-f004:**
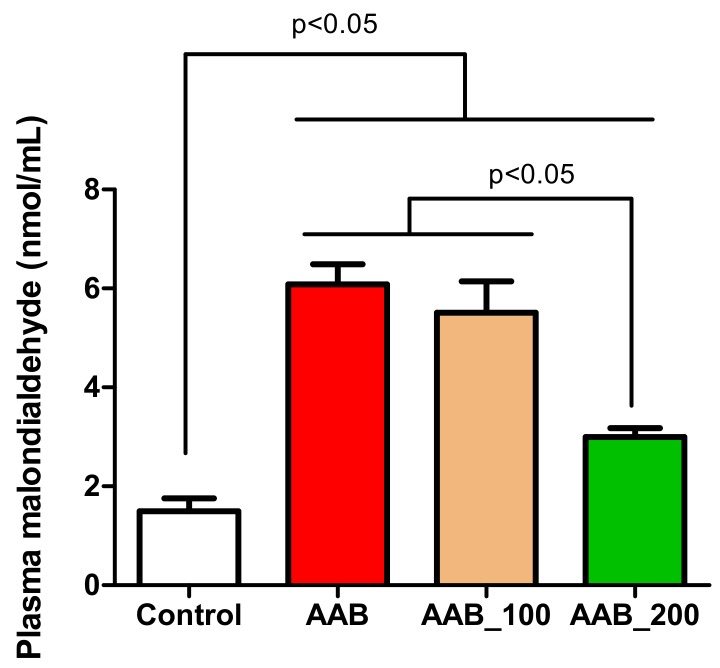
Effects of LBPs on plasma malondialdehyde levels at week 36.

**Table 1 molecules-25-00466-t001:** Morphometric parameters at 36 weeks post-surgery.

Organ/BW Ratio (mg/g)	Control (*n* = 10)	AAB (*n* = 12)	AAB_100 (*n* = 7)	AAB_200 (*n* = 9)
Heart/BW	1.59 ± 0.21	2.28 ± 0.33 *	2.27 ± 0.18 *	2.12 ± 0.18 *
Lung/BW	4.70 ± 0.24	5.03 ± 0.63	4.87 ± 0.47	4.75 ± 0.66
Kidney/BW	1.96 ± 0.29	2.74 ± 0.41 *	2.96 ± 0.42 *	2.69 ± 0.35 *
Liver/BW	23.02 ± 1.90	23.57 ± 3.00	23.55 ± 3.86	23.44 ± 2.12

BW—body weight. * *p* < 0.05 vs. control.

**Table 2 molecules-25-00466-t002:** Effects of LBPs on hematology parameters.

Hematology Parameters	Control (*n* = 10)	AAB (*n* = 12)	AAB_100 (*n* = 7)	AAB_200 (*n* = 9)
WBC	9.01 ± 2.94	9.46 ± 1.55	9.10 ± 2.82	9.18 ± 2.65
LYM	6.06 ± 1.62	6.50 ± 1.44	5.89 ± 1.63	6.57 ± 2.55
MID	0.66 ± 0.52	0.61 ± 0.35	0.59 ± 0.35	0.44 ± 0.13
GRA	2.29 ± 1.04	2.34 ± 0.72	2.63 ± 1.81	2.17 ± 0.81
LY	69.16 ± 9.35	68.09 ± 8.29	66.67 ± 12.08	70.49 ± 8.72
MI	6.56 ± 4.44	6.43 ± 3.21	5.89 ± 2.29	5.06 ± 2.10
GR	24.27 ± 7.06	25.46 ± 8.59	27.49 ± 11.14	24.41 ± 8.74
RBC	8.50 ± 0.63	8.62 ± 0.42	8.14 ± 0.57	8.14 ± 0.39
HGB	14.74 ± 0.96	14.89 ± 0.52	14.70 ± 0.33	14.45 ± 0.27
HCT	42.35 ± 2.37	42.80 ± 2.09	45.85 ± 4.07	45.18 ± 2.20
MCV	50.11 ± 2.57	49.67 ± 1.57	56.29 ± 2.06	55.67 ± 1.73
MCH	17.38 ± 0.89	17.54 ± 0.66	17.59 ± 0.54	17.13 ± 0.39
MCHC	34.77 ± 0.50	35.33 ± 0.68	31.26 ± 0.83	30.83 ± 0.75
RDWc	15.50 ± 0.74	16.61 ± 0.95	15.23 ± 1.05	15.52 ± 0.87
PLT	672.60 ± 34.04	636.68 ± 195.5	673.75 ± 103.35	622.17 ± 103.34
PCT	0.40 ± 0.23	0.49 ± 0.16	0.53 ± 0.27	0.37 ± 0.13
MPV	6.86 ± 0.46	6.96 ± 0.30	7.24 ± 0.63	7.02 ± 0.34
PDWc	33.76 ± 2.61	33.17 ± 0.93	33.70 ± 2.14	32.94 ± 0.87

WBC—white blood cells count (10^3/µL), LYM—lymphocytes count (10^3/µL), MID—other leucocytes (except lymphocytes and granulocytes) count (10^3/µL), GRA—granulocytes count (10^3/µL), LY—lymphocytes percentage (%), MI—other leucocytes percentage (%), GR—granulocytes percentage (%), RBC—red blood cells count (10^6/µL), HGB—hemoglobin (g/dL), HCT—hematocrit (%), MCV—mean corpuscular volume (fL), MCH—mean corpuscular hemoglobin (pg), MCHC—mean corpuscular hemoglobin concentration (g/dL), RDWc—red cell distribution width (%), PLT—platelets (10^3/µL), PCT—platelet proportion in plasma (%), MPV—mean platelet volume (fL), PDWc—latelet distribution width (%). Values are presented as mean ±SD. Student *t*-test, *p* ≥ 0.05 control vs. AAB, AAB vs. AAB_100 and AAB vs. AAB_200.

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
