# Peer review of "Effects of Lycium barbarum L. Polysaccharides on Inflammation and Oxidative Stress Markers in a Pressure Overload-Induced Heart Failure Rat Model"

_molecules, 2020, doi:10.3390/molecules25030466_

Round 1

Reviewer 1 Report

Dear authors, the manuscript entitled "Effects of Lycium barbarum polysaccharides on 2 inflammation and oxidative stress markers in a 3 pressure overload-induced heart failure rat model" deals with a very and prement subject that affects our society well.being. it is well writen and organized, just need minor revisions (find in the document attached). My recommendation is to be published in Molecules journal. 

Author Response

The authors thank the Reviewer 1 for the careful review of our manuscript and for the interesting suggestions he/she made.

Reviewer’ comments

Point 1: “Place botanical marker for Lycium barbarum” (title)

Response 1: As suggested, we have placed the botanical marker “L.” after Lycium barbarum, in the title, abstract (line 28), introduction (line 75) and discussion (line 272). Also, we abreviated “Lycium barbarum L” as “L. barbarum” (lines 75, 272). Also, the old abbreviation (LB) was replaced with “L. barbarum”(line 79).

Point 2: “in the keywords, replace Lycium barbarum with L. barbarum”

Response 2: As suggested, we have done this replacement at line 41 (introduction) and line 284 (discussion).

Point 3: “place a reference after the definition of heart failure in the introduction (line 42).”

Response 3: As suggested, we have placed the reference [1] (Ponikowski, P.; Voors, A.A.; Anker, S.D.; Bueno, H.; Cleland, J.G.F.; Coats, A.J.S.; Falk, V.; 361 González-Juanatey, J.R.; Harjola, V.-P.; Jankowska, E.A.; et al. 2016 ESC Guidelines for the diagnosis 362 and treatment of acute and chronic heart failure. Eur. Heart J. 2016, 37, 2129–2200.) after the the definition and calssification of heart failure in the introduction (line 45).

Point 4: “very long introduction and a little bit confusing with all the acronyms used”

Response 3: As suggested, we have tried to decrease the number of acronyms used and deleted “ROS” at line 71, as it is not used further.

Point 5: “Please say the main methodology used”

Resonse 5: As suggested, at line 79, we added “by echocardiography and analysis of inflammation and oxidative stress markers,” and to avoid repetitions we deleted “on inflammation and oxidative stress markers”.

Point 6: “Figure 1 lacks the legend at x axis”

Response 6: As suggested, we better positioned Figure 1 at lines 110-122 so that the legend for x axis would be visible.

Point 7: “Remove the mg/g from the table”

Response 7: As suggested, we removed “mg/g” and placed it at the header of the table, beneath “Organ/BW ratio”.

Point 8: “Please review the significative numbers” (table 1)

Response 8: As suggested, we have reviewed the statistics and indeed the significance is p<0.05. So we replaced “p<0.01” with “p<0.05” in the footnote of the table (line 133) and in the text (line 124).

Point 9: “Remove the units from the body table 2 and place them in the footnote”

Response 9: As suggested, we have removed the units from the body table 2 and placed them in the footnote.

Point 10: “Please review the significative numbers” (table 2)

Response 10: As suggested, we have reviewed the statistics and indeed there is no significant difference between grous regarding hematology parameters.

Point 11:  “Place botanical marker for Lycium barbarum

Response 11: As suggested, we have placed the botanical marker “L.” after Lycium barbarum at line 284.

Reviewer 2 Report

In this paper, the authors have evaluated the effect of Lycium barbarum polysaccharides on cardiac remodelling and on inflammation and oxidative stress markers in a pressure overload-induced heart failure rat model induced by abdominal aortic banding. The could successfully induce heart failure with moderately reduced ejection fraction and demonstrate beneficial effect of L.b. polysaccharides on the inflammation markers, while there was no measurable affect on cardiac remodelling (heart weight, echo parameters and contractility). There results are clearly presented but there are some flaws in the text, as outlines below. Possible points to improve:

Abstract: "After 12 weeks of treatment with LBPs, the decline of cardiac function was not as important compared to control AAB rats." it is absolutely unclear and confusing what "not as important" means

2. Results "Evolution of body weight" - evolution is not a right word to describe what is meant to be said.

3. Results page 3: "ligatured rat groups compared to control group (Table 1), suggesting cardiac enlargement due to hypertrophy and dilation." Ligatured rat groups sound awkward. Increase in HW/BW ratio is not indicative of dilation

4. Line 112: "Echocardiographic modifications"  sounds awkward

5. Line 264: "needs further investigating." sounds awkward

6. Line 281: "At the end of the treatment period, in the no treatment group (control AAB group) EF, an important marker of systolic function,  decreased even further, whereas in the treatment groups (AAB_100 and AAB_200) EF and LVEDD remained constant, suggesting possible inhibition of further structural and functional decline after LBPs treatment." Which data /figure shows this???From what I can see there is absolute no difference between control and AAB groups in any of the described parameters. 

Author Response

The authors thank the Reviewer 2 for the careful review of our manuscript and for the interesting suggestions he/she made.

Reviewer’ comments

Point 1: “Abstract: "After 12 weeks of treatment with LBPs, the decline of cardiac function was not as important compared to control AAB rats." it is absolutely unclear and confusing what "not as important" means”

Response 1: As suggested, we have replaced “not as important” with “prevented” (line 35).

Point 2: “Results "Evolution of body weight" - evolution is not a right word to describe what is meant to be said.

Response 2: As suggested, we have deleted “Evolution”, the subtitle states now “Body weight”. (line 88)

Point 3: “Results page 3: "ligatured rat groups compared to control group (Table 1), suggesting cardiac enlargement due to hypertrophy and dilation." Ligatured rat groups sound awkward. Increase in HW/BW ratio is not indicative of dilation”

Response 3: As suggested, we have replaced “ligaturated rat groups” with “AAB rat groups” at line 125 and we have deleted “ enlargement due to” and “and dilation” at line 125.

Point 4: “Line 112: "Echocardiographic modifications" sounds awkward”

Response 3: As suggested, we have replaced "Echocardiographic modifications" with “Modifications in the echocardiographic parameters” (lines 141-142).

Point 5: “Line 264: "needs further investigating." sounds awkward”

Resonse 5: As suggested, we have replaced "needs further investigating." with “and the specific pharmacologic mechanisms involved should be further investigated.” (lines 365-366).

Point 6: “Line 281: "At the end of the treatment period, in the no treatment group (control AAB group) EF, an important marker of systolic function,  decreased even further, whereas in the treatment groups (AAB_100 and AAB_200) EF and LVEDD remained constant, suggesting possible inhibition of further structural and functional decline after LBPs treatment." Which data /figure shows this???From what I can see there is absolute no difference between control and AAB groups in any of the described parameters.”

Response 6: As suggested, we modified the frase, the current form being: “At the end of the treatment period (week 36), in the no treatment group (control AAB group) EF, an important marker of systolic function, decreased with ~10% compared to week 24, whereas in the treatment groups (AAB_100 and AAB_200) EF remained constant, suggesting possible prevention of cardiac structural and functional alteration.” (lines 308-312).

Reviewer 3 Report

In the present study, the effects of Lycium barbarum polysaccharides (LBPs) on inflammation and oxidative stress markers were tested in an experimental model of HFmrEF. The study has a great interest. However, this could be greater if some information about the structure, type, ...., of the polysaccharides will be included. It will be enough with some structure or with a more specific description.

Author Response

The authors thank the Reviewer 3 for the careful review of our manuscript and for the interesting suggestions he/she made.

Reviewer’ comment: “However, this could be greater if some information about the structure, type, ...., of the polysaccharides will be included. It will be enough with some structure or with a more specific description.”

Response : A little reference regarding the composition of the active polysaccharide complex  from L. barbarum has been mentioned in the introduction (lines 79-81). However, as suggested, we have included “Polysaccharides extracted from L. barbarum comprise 6 sugars (galactose, rhamnose, glucose, mannose, arabinose and xylose, molar ratios 2.43, 4.22, 1.38, 0.95, 1 and 0.38, respectively), with furan and pyran rings, alpha and beta anomeric configurations.” In the discussion part (lines 285-287). Also, we included the paper where we extracted this information from, as reference [13] (Masci, A.; Carradori, S.; Casadei, M.A.; Paolicelli, P.; Petralito, S.; Ragno, R.; Cesa, S. Lycium barbarum polysaccharides: Extraction, purification, structural characterisation and evidence about hypoglycaemic and hypolipidaemic effects. A review. Food Chem. 2018, 254, 377–389.).